# What Determinants Are Related to Milk and Dairy Product Consumption Frequency among Children Aged 10–12 Years in Poland? Nationwide Cross-Sectional Study

**DOI:** 10.3390/nu16162654

**Published:** 2024-08-11

**Authors:** Jadwiga Hamulka, Ewa Czarniecka-Skubina, Magdalena Górnicka, Jerzy Gębski, Teresa Leszczyńska, Krystyna Gutkowska

**Affiliations:** 1Department of Human Nutrition, Institute of Human Nutrition Sciences, Warsaw University of Life Sciences (SGGW-WULS), 166 Nowoursynowska St., 02-787 Warsaw, Poland; jadwiga_hamulka@sggw.edu.pl (J.H.); magdalena_gornicka@sggw.edu.pl (M.G.); 2Department of Food Gastronomy and Food Hygiene, Institute of Human Nutrition Sciences, Warsaw University of Life Sciences (WULS), 166 Nowoursynowska St., 02-787 Warsaw, Poland; 3Department of Food Market and Consumer Research, Institute of Human Nutrition Sciences, Warsaw University of Life Sciences (SGGW-WULS), 166 Nowoursynowska St., 02-787 Warsaw, Poland; jerzy_gebski@sggw.edu.pl (J.G.); krystyna_gutkowska@sggw.edu.pl (K.G.); 4Department of Human Nutrition and Dietetics, Faculty of Food Technology, University of Agriculture in Krakow, 31-120 Kraków, Poland; teresa.leszczynska@urk.edu.pl

**Keywords:** milk and dairy products, consumption, nutrition knowledge, schoolchildren, parents

## Abstract

Due to their high nutritional value, milk and dairy products should be a permanent element of a properly balanced diet for children and adolescents. The study aimed to identify (i) the frequency of milk and dairy product consumption by children aged 10–12 years in the opinion of children and their parents and (ii) the determinants related to the consumption of these products (including lifestyle, nutrition knowledge, and the nutritional status of children’s as well as parents’ nutrition knowledge). A cross-sectional study was conducted with 12,643 primary school students aged 10–12 and 7363 parents. Dietary data were collected using the Food Frequency Consumption and Nutritional Knowledge Questionnaire (SF-FFQ4PolishChildren^®^ and KomPAN^®^). Anthropometric measurements were taken and body mass index (BMI) and waist/height ratios (WHtR) were calculated. A logistic regression model was used to assess the likelihood of the frequent consumption of dairy products in the opinion of both the children and their parents, and the quality of the obtained models was assessed using model fit statistics and the Hosmer and Lemeshow test. The frequency of consuming milk and milk products (every day and more) was low in the opinion of children aged 10–12 years (29.6%). According to their parents, the frequency of the consumption of dairy products was slightly higher (44.8%). Six factors associated with the frequency of milk and dairy product consumption were identified independently. These included schoolchildren and their parents’ nutrition knowledge, physical activity, sleep, gender, and place of residence. More of these products were consumed by children with greater nutrition knowledge—both their own and their parents’—higher physical activity, longer sleep duration, males, and those living in the city. Family eating habits, particularly eating meals together, also played an important role in the consumption of milk and dairy products. According to the parents, the children in older grades were less likely to consume dairy products more frequently. The obtained results suggest an insufficient consumption of milk and dairy products. The identification of modifiable factors, such as nutrition knowledge, physical activity, sleep duration, and eating meals with the family, suggests the need to improve the existing strategies, including activities encouraging nutritional education for both schoolchildren and their parents.

## 1. Introduction

Milk and dairy products, due to their high nutritional value and digestibility, as well as effective absorption of nutrients, should be permanent elements of diet, especially among schoolchildren [1]. The period of development and rapid growth in a young organism requires the daily consumption of these products due to their high content of calcium, as well as easily digestible, high-quality protein used to build cells, tissues, and organs. Calcium is a building component of bones and teeth, a key factor in blood clotting, a regulator of the activity of the muscular and nervous systems, and an enzyme activator. Milk, as the main source of calcium in the diet, covers over 50% of its total intake [1,2]. Milk fat is also an important component of milk and dairy products due to vitamins D and A, conjugated linoleic acid (CLA), coenzyme Q10, and phospholipids, with antioxidant and emulsifying properties. For this reason, milk fat is the only one that does not need bile acids for digestion and is recommended in dietetics. Milk is also the basic source of vitamin B_2_, B_12_, and phosphorus. Therefore, the inclusion of these products in the daily diet is very important [2,3,4,5].

Dairy products have played an important role in the human diet for approximately 8000 years and are part of the official dietary recommendations in many countries around the world [6,7,8,9,10]. Generally, in European countries it is recommended to consume 2–3 portions of milk and milk products, preferably low-fat ones, and without added sugars [6]. They provide crucial nutrients, which is particularly important for school-aged children due to their nutritional requirements related to rapid growth and maturation [5,8,9,10]. Milk consumption recommendations for children vary according to age and specific nutritional needs. According to the American Academy of Pediatrics (AAP) [11], school-aged children (up to 8 years of age) should consume 2–2.5 servings of milk per day, equivalent to 450–600 mL. Children aged 9–18 are recommended to consume three daily portions of low-fat (1% fat) or skimmed milk, which corresponds to approximately 700–900 mL of milk.

Despite its high nutritional value, milk is one of the most controversial food products and there are many myths surrounding its consumption. Myths about milk often result from an overinterpretation or misunderstanding of scientific research results [3,12]. Study results clearly indicate that milk and yogurt consumption is associated with healthy growth in children’s height and weight without increasing the risk of obesity [1,13,14,15]. However, as shown by Yasar et al. [15], insufficient milk consumption was associated with higher waist circumference in girls, which may be associated with abdominal obesity and, in the future, the development of chronic non-communicable diseases associated with obesity.

Milk and its products have been a particularly valuable food in the average Polish diet for centuries due to their high nutritional density and the good bioavailability of the components of milk and its products. The total share of calcium from dairy products in the Polish diet is 55%. Therefore, the role of milk and its products in providing calcium cannot be overestimated, especially since its average consumption in Poland is low and amounts to approximately 60% of the recommended dose. In addition, milk in the Polish diet is important in providing riboflavin and vitamin B12 [9].

Milk may also be a valuable source of iodine, considering the relationship between iodine deficiency and brain health or the impact on cognitive functions [16,17].

It is also worth emphasizing that the consumption of dairy products, especially cheese, and yogurt is associated with a reduction in dental caries in children which is a common problem in the pediatric population in Poland. Moreover, the results of two prospective cohort studies support an inverse association between dairy consumption in early childhood and blood pressure in middle childhood or early adolescence. The evidence so far suggests that dairy products are important for linear growth and overall health, not just for bones [1].

The taste of milk and natural dairy products may not be acceptable to some children and may limit their consumption. However, it seems that the early introduction of cow’s milk into a child’s diet and its subsequent regular consumption can shape positive eating habits in this regard. Children’s preference for sweet tastes, however, may influence their choice towards dairy products with an increased sugar content, such as sweet dairy desserts or flavored milk. This may contribute to greater sugar intake in the diet, including from drinks, which may lead to overweight and obesity and subsequently, non-communicable chronic diseases [18,19].

Unfortunately, over the last two decades, a decline in milk consumption has been observed in Poland, from 64.68 L per person per year in 2000 to 34.2 L per person per year in 2022. In contrast, the consumption of dairy products such as yogurts, cheese, cottage cheese, and sour and sweet cream has remained constant, from 22.2 L per person/year to 22.32 l per person/year. For this reason, it is important to understand the factors influencing the consumption of milk and milk products, especially among adolescents [20,21].

Given this, this study aimed to investigate (i) the frequency of milk and dairy product consumption by children aged 10–12 years in the opinion of children and their parents and (ii) the determinants related to the consumption of these products (including lifestyle, nutrition knowledge, and the nutritional status of children’s as well as parents’ nutrition knowledge).

## 2. Materials and Methods

### 2.1. Ethics Approval

The project followed the ethical standards recognized by the Helsinki Declaration. The study protocol was also approved by the Ethics Committee of the Institute of Human Nutrition Sciences of the Warsaw University of Life Sciences (Resolution No. 18/2022). Written informed consent was obtained from parents for their children’s participation.

### 2.2. Study Design and Participants

The analyses related to this article were designed as a cross-sectional study as a part of the Junior-Edu-Żywienie (JEŻ) Nationwide Project. We used a two-stage cluster sampling design to select a representative sample of students aged 10–12 years from public and private schools. The invitation to participate in the project/research was sent to all the primary schools (*n* = 14,322) in Poland, but 2208 schools responded positively. The cluster sampling design consisted of (1) the first stage involving selection reported by school principals with a “probability proportional to macroregion and the size of the place”, and (2) the second stage involving randomly selecting classes within each school. In the selected classes, all the students were eligible to participate in the survey, but not all parents consented. The data were collected in 2022–2023 by well-trained university researchers. Ultimately, children were recruited from 140 primary schools, targeting those aged 10–12 years from the entire territory of Poland. More details on the study design, methods, and sample collection have been described in a previous paper [22] and are presented in a flowchart of this study (Figure 1). Ultimately, the study took 12,643 children aged 10–12 years, comprising 6423 boys (50.8%) and 6220 girls (49.2%), as well as 7363 parents. There were 6668 pairs from the same family.

### 2.3. Data Collection

To collect data related to diet, including the frequency of the consumption of milk and dairy products as well as lifestyle factors such as physical activity, screen time, and sleep time, in addition to nutrition knowledge and sociodemographic factors, i.e., gender, age, and place of residence, a validated tool dedicated to Polish teenagers, SF-FFQ4PolishChildren^®^, was used [23], supplemented with questions about meals shared with the family (how often do you eat (various) meal together with family?). In a paper-and-pencil format, the questionnaire was self-administered by the schoolchildren in classrooms under the supervision of researchers and teachers. More details on the study protocol and methods were described previously [22].

#### 2.3.1. Children—Milk and Dairy Products Frequency

The participants reported the habitual frequency of consuming milk and dairy products within the previous 12 months by selecting one of the following answers: 1—never or almost never; 2—less than once a week; 3—once a week; 4—two–four times a week, 5 —five-six times a week; 6—daily; and 7—several times a day. During the data analysis, these values were converted into two categories: low frequency (max. 2–4 times a week) and high frequency (at least 5–6 times a week). Additionally, the results were analyzed using a division consistent with the recommendation to consume milk and dairy products every day (“every day and more” and “less than every day”).

#### 2.3.2. Children—Covariates

Screen time (ST) was determined through the following question: “how much time do you spend in front of a computer, tablet, phone, or TV on an average day of the week?” The recommendation of the American Academy of Pediatrics, which suggests a maximum of two hours of ST per day for children and youth, was used as a reference [24,25]. The participants reported their screen time by providing one of the following answers: <2 h/day; 2 to <4 h/day; 4 to <6 h/day; 6 to <8 h/day; 8 to <10 h/day; and ≥10 h/day. Due to the small number of respondents in higher categories, three ST categories were considered in further analysis: <2 h/day, 2–4 h/day, and >4 h/day.

Sleep time was assessed using the question “how many hours do you spend sleeping on an average weekday?” with three answer options: “less than 6 h/day”, “from 6 to almost 8 h/day”, and “8 or more hours/day”.

Physical activity (PA) during leisure time was assessed using the question “how would you describe your physical activity during leisure time/your time off (after classes and on weekends)”? The respondents chose one of three categories: low, moderate, or vigorous. Examples for each category of PA were provided. The following response categories for PA were used: low (more time spent sitting, watching TV, in front of a computer, reading, light housework, or a short walk to 2 h a week), moderate (walking, cycling, gymnastics, working at home, or other light physical activity performed 2–3 h/week), or vigorous (cycling, running, working at home, or other sports activities requiring physical effort over 3 h/week).

Additionally, the schoolchildren were asked about the frequency of eating meals with their family using the question “how often do you eat meals together with your family?” The frequency of eating meals with their family was assessed by selecting one of the following response categories: not at all; less than 1 time/week; 1–2 days/week; 2–4 days/week; 5–6 days/week; or every day. Explanations were provided if necessary. This questionnaire was described in the study protocol [22].

In our analysis, family economic status was not considered because milk and dairy products are cheap in Poland and this is not a factor determining their consumption. This was discussed in more detail in the limitations.

#### 2.3.3. Parents

The study was also conducted among the parents, who completed another questionnaire with content similar to the questionnaire for the schoolchildren aged 10–12 years. The questionnaire contained the same FFQ from the KomPAN^®^ questionnaire [26,27] (covering the same food groups and frequencies of consumption) and three questions about the child’s lifestyle, including physical activity, screen time, and sleep time as well as about shared meals with family/children.

### 2.4. Nutrition Knowledge Score

The assessment of general nutrition knowledge was based on 20 closed questions. Five of these questions were taken from the validated questionnaire dedicated to Polish youth, SF-FFQ4PolishChildren^®^ [23]. The remaining 15 questions referred to the current dietary guidelines for Polish children and adolescents [28].

A correct answer was scored with 1 point, while wrong or “I don’t know” answers, as well as missing data, were scored with 0 points. The points were summed for each respondent (range 0 to 20 points). Based on tertile distribution, the respondents were divided into nutrition knowledge categories: low (0–7 points), medium (8–14 points), and high (15–20 points).

Regarding knowledge about milk and its products, in the knowledge test, both the children and parents had to choose one answer to the following question: the main sources of calcium in the diet are (i) milk and dairy products; (ii) vegetables and fruits; (iii) dairy products and meats; (iv) “I don’t know”. The correct answer (i) was classified as “proper”, while the remaining answers were classified as “improper”.

### 2.5. Anthropometrics

Anthropometric parameters: body height (H), body weight (BW), and waist circumference (WC) were measured using the standardized procedures and equipment described previously [22]. Body mass index (BMI, kg/m^2^) and waist/height ratio (WHtR) were calculated. BMI-for-age was categorized according to the Polish sex-specific cut-offs for children [29] as follows: underweight BMI < 18.5 kg/m^2^; normal weight BMI = 18.5 to 24.9 kg/m^2^; and overweight BMI ≥ 25 kg/m^2^ (general adiposity marker), and WHtR ≥ 0.5 as a central obesity measure (central adiposity marker) [30].

### 2.6. Statistical Analysis

A preliminary analysis of the obtained results was carried out. Due to the qualitative nature of the variables analyzed, frequency tables and multi-way tables were used for their assessment. To assess the likelihood of the frequent consumption of dairy products, both in the opinion of the schoolchildren and their parents, a dependent variable of the model was selected, describing the rare (at most 2–3 times a week) or frequent (at least 5–6 times a week) consumption of dairy products. Due to the dichotomous nature of the dependent variable (regressand) (the rare/frequent consumption of dairy products), a logistic regression model was used. Independent variables (regressors) were also determined to explain the dependent variable: nutrition knowledge (categorized), physical activity, sleep duration, gender, age (class), the frequency of eating meals together with the family, the size of the place of residence, and BMI and waist/height ratio (WHtR) as markers general adiposity and central adiposity, respectively. Two models were developed for comparison, describing the same relationship. The first model concerned the opinion of the children about their own consumption of dairy products, while the analogous model concerned the opinion of the parents about their children’s consumption of these products. The impact of individual explanatory variables on the dependent variable was assessed based on odds ratios (ORs) and 95% confidence intervals (CIs).

The fit statistical model and the Hosmer and Lemeshow test were used to assess the quality of the obtained models. All the statistical analyses were conducted with SAS 9.4. For all the tests, *p* ≤ 0.05 was considered significant. Differences between the groups were verified using the chi-square test. The predicted level of the dependent variable in both the models was “frequent—at least 5–6 times a week” consumption of dairy products.

## 3. Results

### 3.1. Characteristics of the Study Sample

The sociodemographic descriptions of the studied group of schoolchildren are presented in Table 1. Over 52% of the respondents reported consuming milk and milk products at a low frequency (max. 2–4 times a week), with a slightly higher proportion of girls than boys (50.3 vs. 49.7%). Overweight was found in 15.3% of the children aged 10–12 years, and underweight in 11.4% of the participants, but no differences were noted taking into account the gender of the schoolchildren studied, while central obesity, identified as a waist/height ratio (WHtR) ≥ 0.5, was found in 16.8% of the children, including 18.2% of the boys and 13.6% of the girls (Table 1).

### 3.2. Frequency of Consumption of Milk and Dairy Products Among Schoolchildren in the Opinion of Parents and Children

It was shown that, according to the schoolchildren’s self-report, approximately 30% of them consumed milk and dairy products every day and several times a day, but in the opinion of their parents, this figure was 45% for both in the total group and the “paired” group. Statistically significant differences were observed in the assessment of the frequency of the consumption of milk and dairy products (Table 2). According to the parents, their children consumed these products more frequently than what their children themselves declared. The children and parents did not agree in their answers in this respect. Every day and more was agreed upon by only 18% of the pairs of children and their parents, and less than every day by 42.9%.

The frequency of milk and dairy product consumption by the children was not related to the children’s BMI category (Table 3) both in the total group and in the division by gender of the children studied (*p* > 0.05). These relations were also not shown for the WHtR index (unpublished data)

### 3.3. General and Specific Nutrition Knowledge of Parents and Children

A statistically significant difference was found in the level of general nutrition knowledge between the children and parents, both in the entire group and in the “paired” group (Table 4).

A high level of total nutrition knowledge was observed in almost 68% of the parents compared to 8–14% of the schoolchildren. On the other hand, a high level of nutrition knowledge about milk and milk products was noted in almost 90% of the parents compared to 44% of the children. It was also noted that nutrition knowledge varied by the gender of the parent, with slightly more correct answers given by women.

Even though the parents had, for example, high general nutrition knowledge (67.9%), particularly about milk and dairy products (88.7%), the consistency of the answers of the children and their parents after pairing was only 11.3% and 39.8%, respectively. This indicates the importance of knowledge in the field of nutrition acquired at school when parents do not have this knowledge.

### 3.4. A Model of the Influence of Schoolchildren’s Nutrition Knowledge and Selected Lifestyle Elements on the Frequency of Their Consumption of Milk and Dairy Products

Average nutrition knowledge (8–14 correct answers) was associated with a 21.9% greater likelihood of consuming milk and dairy products more frequently compared to the individuals with low nutrition knowledge (0–7 correct answers) (odds ratio OR: 1.22; 95% confidence intervals CI: 1.12–1.32). The schoolchildren with a high level of nutrition knowledge (15–20 correct answers) showed a 66.5% greater likelihood of consuming dairy products more frequently compared to the reference level (individuals with low nutrition knowledge) (OR: 1.65; 95% CI: 1.44–1.92), with the remaining model parameters held constant (Table 5).

In the case of the schoolchildren’s subjective assessment of their own nutrition, an increase in this assessment by each point increased the likelihood of consuming dairy products more frequently by 13.5% (OR: 1.14; 95% CI: 1.08–1.19).

The children aged 10–12 years who declared average levels of physical activity had a 9% greater likelihood of consuming dairy products compared to the students who reported low physical activity (OR: 1.09; 95% CI: 0.96–1.22), although this result was not statistically significant in the model. However, the adolescents who reported being very active physically had a 26% greater likelihood of consuming dairy products more frequently compared to those who were not physically active (OR: 1.26; 95% CI: 1.11–1.42).

Sleep duration had a limiting effect on the dependent variable. The children aged 10–12 years who slept less than 6 h per day had a 25% less likelihood of consuming dairy products more frequently compared to those sleeping 8 h or more per day (OR: 0.75; 95% CI: 0.66–0.85). Conversely, the people who slept between 6 and 8 h a day had a 19% lower likelihood of consuming dairy products more frequently than those who slept 8 h or more per day (OR: 0.81; 95% CI: 0.65–0.87).

Regarding the gender of the respondents, it was found that the girls had a 7% lower likelihood of consuming milk and dairy products more frequently than the boys (OR: 0.93; 95% CI: 0.87–0.99).

In turn, age did not show statistically significant differences in the frequency of the consumption of milk and its products.

The place of residence also influenced the frequency of milk and dairy product consumption among the schoolchildren in grades 4–6. Urban residents were 9% more likely to consume dairy products more frequently compared to rural residents (OR: 1.08; 95% CI: 1.04–1.18).

The variable related to the frequency of eating meals with the family was significant in the model, though only one of its levels proved to be statistically significant. The children who ate meals with their family every day were 43% more likely to consume dairy products more frequently compared to those who never ate meals with their family (OR: 1.43; 95% CI: 1.16–1.76).

### 3.5. A Model of the Influence of Parents’ Nutrition Knowledge and Assessment of Selected Elements of Children’s Lifestyle on the Frequency of Consumption of Milk and Dairy Products by Children

Average nutrition knowledge among the parents (8–14 correct answers) was associated with a 49% greater likelihood of more frequent consumption of milk and dairy products by their children (Table 6) compared to the parents with a low level of nutrition knowledge (0–7 correct answers) (OR: 1.49; 95% CI: 1.12–1.97). The children of the parents with a high level of nutrition knowledge (15–20 correct answers) had a 100% greater chance of consuming milk and dairy products more frequently compared to the reference level (parents with low nutrition knowledge) (OR: 2.00; 95% CI: 1.52–2.64), with the remaining model parameters kept constant.

In the case of the parents’ subjective assessment of the children’s nutrition, an increase in this assessment by each point increased the chance of the children consuming dairy products more frequently by 21% (OR: 1.21; 95% CI: 1.13–1.29).

The parents’ assessment of the level of the children’s physical activity did not result in differences in the chance of the children consuming milk and dairy products more frequently (the variable was statistically insignificant in the model).

Shorter sleep duration had a limiting effect on the dependent variable. The schoolchildren who slept less than 6 h per day had their likelihood of consuming dairy products more frequently reduced by half compared to those who slept 8 h or more per day (OR: 0.51; 95% CI: 0.27–0.95).

Additionally, the children who slept 6–8 h a day had a 23% lower likelihood of consuming milk and dairy products more frequently compared to those who slept 8 h or more per day (OR: 0.77; 95% CI: 0.69–0.85).

The girls had a 7% lower chance of consuming dairy products more frequently compared to the boys (OR: 0.93; 95% CI: 0.84–0.98), which is consistent with the results obtained from the children’s self-assessment.

With age, the likelihood of consuming milk and dairy products more frequently decreased. The schoolchildren of older grades, in the opinion of their parents, had a lower chance of consuming dairy products more frequently compared to the fourth-grade schoolchildren. This likelihood decreased by 20% for the fifth-grade adolescents (OR: 0.80; 95% CI: 0.71–0.89) and 14% for the sixth-grade adolescents (OR: 0.86; 95% CI: 0.76–0.97).

The place of residence influenced the likelihood of consuming milk and dairy products according to both the parents and children. City residents in areas with a population of more than 500,000 residents had a 38% greater chance of consuming milk and dairy products more frequently compared to the children living in rural areas (OR: 1.38; 95% CI: 1.16–1.65).

According to the variable regarding the frequency of eating meals with the family, the parents’ opinions indicated that those who ate meals with their family 3–4 times a week were 28% more likely to consume dairy products more frequently compared to those who ate meals with their family less than once a week (OR: 1.28; 95% CI: 1.09–1.78).

The school-aged children who ate meals 5–6 days a week with their family had a 43% greater likelihood of consuming dairy products more frequently compared to those who ate meals with their family less than once a week (OR: 1.43; 95% CI: 1.02–2.01). Moreover, in the people who ate meals every day with their family, this likelihood increased by 54% compared to those who had family meals less than once a week (OR: 1.54; 95% CI: 1.10–2.14).

## 4. Discussion

According to the study criteria, the frequency of milk and dairy product consumption was found to be insufficient in most adolescents aged 10–12. Milk consumption by children and adolescents is influenced by various interacting factors. Our study successfully identified six factors independently associated with a higher frequency of milk and dairy product consumption: the nutrition knowledge of students and their parents, higher physical activity, longer sleep duration, male gender, and living in an urban area. Interestingly, family eating habits, particularly eating meals together, positively influenced habits related to the consumption of milk and dairy products. Additionally, in parents’ opinion, older children were less likely to consume dairy products more frequently.

Milk and dairy products possess high nutritional value and are recommended as part of a balanced diet. Despite their importance in the diet of children and adolescents, data indicates a continuous decline in their consumption with age [31]. The results obtained in this study align with nationwide data [32] regarding the diet and nutritional status of the Polish population, indicating that only 22.2% of the boys and 26.4% of the girls aged 10–17 consumed milk at least once a day. A large group of adolescents declared drinking milk 2–3 times (36.8% of the boys and 9.3% of the girls) or 4–5 times a week (25% and 22.6%). The most frequently consumed milk drink was yogurt. However, only 7.9% of the boys declared consuming it daily. Low consumption of cheese, especially cottage cheese, was also observed with 1.4–1.7% of the respondents. Low consumption of milk and dairy products (approximately 190 g/day) was also observed in the German population [5]. Moreover, the DONALD study, spanning 1985–2019, involving 1275 participants aged 3.5–18.5 years, and using multinomial mixed regression models, showed changes in the consumption patterns of milk and its products among children and adolescents over the last three decades, demonstrating a decline in their consumption with age with a simultaneous increase in the consumption of fermented milk products. The consumption of low-fat and high-sugar dairy products has also changed over time, and, according to the authors, further assessment is required to examine the health significance of these changes in consumption patterns among children and adolescents [31,32].

The relationship observed in our research between living in a city and more frequent consumption of milk and dairy products is confirmed by the data from the UNICEF report [33]. In general, regardless of geographical region, urban residents have greater access to basic goods and services, including food. Cities are also more closely connected to the supply chain, which undoubtedly translates into greater availability of dairy products for families living in cities.

Children’s milk consumption is shaped by many factors that include biological and social, as well as taste preferences and eating habits [34,35,36]. The factors that may influence these trends over time and within a given population group (cohort) include parental influence, gender, the substitution of milk with other beverages, and overall dietary quality [37,38]. Family eating habits also play a significant role in shaping children’s consumption of milk and dairy products. Parents who regularly consume milk and incorporate it into family meals often pass these habits on to their children [39]. Educational campaigns promoting healthy eating or the School Program present in Polish schools can increase awareness of the benefits of milk consumption. Additionally, peer influence within social environments can impact children’s food preferences, including milk consumption [39,40,41]. A Brazilian study found that insufficient milk intake among teenagers was linked to factors such as a lower frequency of breakfast consumption, lower maternal education levels, a lack of physical activity, and unhealthy eating habits [42,43].

Our results indicated a positive relationship between high physical activity and more frequent consumption of milk and its products. Similarly, Silva et al. [42] found that physical inactivity was associated with insufficient milk intake among adolescents. Physical inactivity is associated with consuming foods of low nutritional quality and high energy density (ultra-processed foods). This lack of physical activity not only affects dietary choices but also contributes to less diverse diets, resulting in a lower intake of essential nutrients. In addition to the inappropriate consumption of milk and dairy products, it is also associated with the reduced consumption of legumes, fruit, meat, vegetables, and cereals.

Similar to the results of our study, the Larson et al. study [44] highlighted that in the group of teenagers, low milk consumption was more prevalent among girls. This is concerning since adequate calcium intake up to and during adolescence is necessary for achieving optimal bone mass and density.

Although our study did not show an association between the frequency of milk and dairy product consumption by children and BMI and WHtR as simple markers of general and central obesity, some studies show a neutral or negative association between milk and dairy product consumption and obesity, but it depends on the type of products consumed (unsweetened vs. sweetened) [1,13,15]. A systematic review and meta-analysis of the associations between dairy intake and adiposity in preschoolers, school-aged children, and adolescents in developed countries found no significant association between dairy intake and adiposity in the aggregated data. However, among adolescents, dairy intake was inversely associated with adiposity (*p* < 0.0001). Despite limitations, these data suggest, however, a neutral effect of dairy intake on adiposity in early and middle childhood and a modest protective effect in adolescence [43]. In addition, Nezami et al. [45] showed that there seems to be a specific gender role of dairy intake in the tendency for central adiposity and body composition in children aged 12–18 years in Southern California and Michigan. They found that total dairy intake was positively associated with WHtR, fat mass, and fat-free mass, but only in boys.

Sufficient calcium intake in girls is not only essential for reproductive health, but also protects them against osteoporosis later in life, particularly in postmenopausal women [46]. It should be emphasized that girls have a generally better overall diet quality [47,48].

Our results can support developing targeted interventions aimed at improving milk and dairy product consumption in the subpopulation of girls.

Recent research has also demonstrated that the dietary intake of milk and dairy products can serve as an indicator of sleep quality (a well-balanced diet that includes milk and dairy products is considered effective in improving sleep quality) [49]. This effect is thought to be related to the large amount of tryptophan (Try) in milk, which is a precursor for the synthesis of serotonin and then melatonin. A cross-sectional study has suggested associations between higher milk intake, improved ease of falling asleep, and an earlier chronotype [50].

The consumption of milk and dairy products is closely related to meal patterns throughout the day [42]. Breakfast, as a basic daily meal, is often associated with the consumption of milk in the Polish tradition. Larson et al. [44] confirmed that the consumption of milk and dairy products was associated with the consumption of breakfast. Skipping breakfast may, therefore, contribute to low milk consumption. In the present study, both the children and parents indicated lunch (77% and 72%, respectively) and dinner (42% and 8%, respectively) as meals eaten together, while breakfast was identified by every fourth child and every fifth parent. This may explain the results obtained regarding the relationship between the frequency of eating family meals and the consumption of milk and dairy products.

The research results confirm the role of the family in shaping behavior in the consumption of milk and its products by adolescents [51]. Moreover, as recent studies have shown, a higher frequency of family meals was associated with a higher intake of many (12/21) nutrients (e.g., protein, fiber, and potassium), which may be related to a more favorable diet. Moreover, a higher consumption of milk and dairy products is associated with a lower consumption of soft drinks and a higher consumption of fruit, vegetables, and cereals in children and adolescents, which correlates to healthier eating habits overall [52,53,54].

Because stable food preferences are best formed from an early age, activities promoting milk consumption can effectively lead to its increase. In Polish primary schools, as in other European schools, a School Program was introduced under which children receive subsidized milk [40]. Undoubtedly, any activity that promotes the increased consumption of milk and dairy products can reduce some of the barriers to their consumption [42]. School authorities (teachers) can also work together to promote a healthy eating environment in their schools to support student health through the distribution of healthy foods [41].

Given the rising prevalence of childhood obesity and studies showing that higher consumption of milk, yogurt, and milk-based drinks (but unsweetened) is associated with lower body fat, a lower risk of cardiovascular disease, and higher cardiorespiratory fitness in adolescents [55,56,57], it is necessary to take effective action in the field of nutritional education and the promotion of this group of products.

### Strengths and Limitations

The strength of our study and its new aspect is the combination of examining nutrition knowledge and dietary habits in the opinion of children and, at the same time, in the opinion of parents in such a large group of the population. The strength of this study is also the large sample size (over 12,000), covering the entire territory of Poland. This diverse group represents the demographic and social spectrum of adolescents aged 10–12, allowing for generalizations to the broader population. The study group reflected the ratio of the rural and small-town to urban population in Poland (40.1/59.9 vs. 40.5/59.5), while the ratio of boys/girls in our sample (50.8/49.2) also mirrored the distribution of Polish youth aged 10–14 (51.2/48.7) [58].

Moreover, the statistical models took into account both sociodemographic factors, i.e., gender, age, place of residence, physical activity in free time, screen time, and sleep duration, as well as the nutrition knowledge of schoolchildren and their parents, in addition to a subjective assessment of diet and nutritional status (BMI and WHtR). The research was conducted among both children and their parents, which allowed us to determine compliance with the consumption of milk and dairy products.

However, it should be noted that this study is not without limitations. First, our study had a cross-sectional study design. It is difficult to establish cause-and-effect relationships using cross-sectional studies because they only provide a single measurement of both the alleged cause and effect. However, cross-sectional studies allow you to collect data from a large number of people and compare differences between groups. This type of study is often used; it has advantages and disadvantages, which should be considered when planning research and interpreting the results [59]. Second, the study used a short FFQ (SF-FFQ4PolishChildren^®^) to assess the consumption of milk and dairy products, which did not allow for an actual assessment of the amount of consumption of these products in the diet. However, the FFQ is a well-established, validated, and widely used questionnaire, and ours was additionally validated on a nationwide sample. Future studies should consider using a long form of the FFQ with a broader list of food items to better describe the usual food consumption and/or applying other methods of lifestyle assessment to fully describe the teenagers’ dietary behaviors. However, to date, there is no validated long form of FFQ which has been developed for Polish children or adolescents. Moreover, the discrepancy between the parents’ and children’s perceptions of consumption frequency could be reduced by using mixed methods, including both self-report and direct observation. Third, in our analysis, family economic status was not taken into account. Based on research analyses and literature review [60,61], it was shown that changes in the level of milk consumption were small in relation to changes in prices expressed in current prices, as well as consumer income. In contrast, there were changes in dairy consumption. Between 2004 and 2020, the consumption of dairy products, i.e., yogurts, cheeses, and cottage cheese, increased, the price of which is higher, while the consumption of liquid milk decreased. Milk and dairy products are relatively cheap compared to other food products in Poland. In addition, economically disadvantaged households receive funds from the program “Family 800+”—support for families to cover a part of the costs of raising a child. In addition, under the “Milk at School” program, financed, inter alia, by the EU and the Milk Promotion Fund, children in grades 1–5 receive milk and milk products free of charge at school. Therefore, analyzing the data in this respect would not be justified. Fourth, due to potential cultural differences in dairy consumption patterns worldwide, it is difficult to generalize our results to other countries, although they can undoubtedly be used in the countries of north-central Europe. Fifth, this study also did not take into account factors related to the assessment of health status and the elimination of milk and dairy products due to lactose intolerance or milk protein allergy. In our study, only 1–3% of the respondents did not consume milk and dairy products. Finally, in view of the observed changes towards replacing dairy products with plant alternatives, further research is necessary to confirm whether this change is common in the group of children and adolescents and whether it will provide them with the appropriate quantity and quality of nutrients such as protein, calcium, or riboflavin.

## 5. Conclusions and Implication

Among Polish teenagers, the frequency of milk and dairy product consumption is inconsistent with the nutritional recommendations. Our research allowed us to identify the subpopulations of children and adolescents at risk of insufficient intake of these products (girls, adolescents living in rural areas, and adolescents with low physical activity). By identifying the factors that determine the frequency of the consumption of milk and dairy products, our findings can contribute to the design of targeted comprehensive, multi-sectoral interventions to increase milk and dairy product consumption in this region. The identification of modifiable factors such as nutrition knowledge, physical activity, sleep duration, and family meal habits may help improve the existing strategies, educational programs, or programs promoting a healthy lifestyle including those aimed at increasing the consumption of milk and dairy products among schoolchildren as an important source of calcium and other nutrients.

It is crucial for nutrition education efforts to encompass both adolescents and their parents, to be implemented both at school and at home, and to comprehensively cover the issues of nutrition, physical activity, and nutrition knowledge.

In addition, further research is necessary to assess the impact of consumption/non-consumption of milk and dairy products on the nutritional status and health of adolescents.

### Our Study Has Several Possible Implications

Future research should use mixed methods that include both self-reports and direct observations to minimize discrepancies between parents’ and children’s perceptions.

Future studies should consider conducting longitudinal studies that can follow the same subjects over time to establish more robust causal relationships between the studied factors and dairy consumption.

Various activities are needed to implement nutrition education for children and adolescents, whole families, and teachers, including education on milk and dairy products. Modern IT and telecommunication tools (e.g., social media and mobile applications) should be used to support traditional methods.

## Figures and Tables

**Figure 1 nutrients-16-02654-f001:**
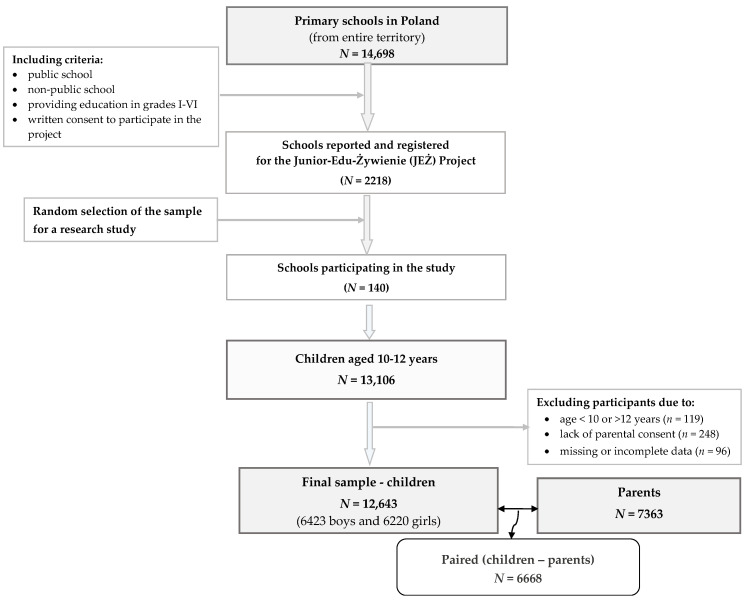
Flowchart of the study design and population.

**Table 1 nutrients-16-02654-t001:** Characteristics of the schoolchildren study group.

Variables	Total Group N = 12,643	Boys N = 6423	Girls N = 6220	*p*-Value *
**Age—years (%):**				
10	38.0 (4799)	48.9 (2349)	51.1 (2450)	0.0004
11	32.0 (4049)	50.7 (2054)	49.3 (1995)	
12	30.0 (3795)	53.2 (2020)	46.8 (1775)	
**Place of living (%):**				
village	21.1 (2699)	50.1 (1337)	49.9 (1332)	0.0117
<20,000 residents	18.0 (2271)	48.3 (1096)	51.7 (1175)	
20,000–100,000 residents	22.9 (2897)	50.5 (1462)	49.5 (1435)	
100,000–500,000 residents	24.4 (3086)	52.8 (1630)	47.2 (1456)	
>500,000 residents	13.6 (1720)	52.2 (898)	47.8 (822)	
**Macroregion:**				
central	24.7 (3177)	50.0 (1558)	50.0 (1559)	0.2645
south-eastern	21.5 (2721)	52.5 (1428)	47.5 (1293)	
south-western	12.3 (1552)	51.6 (801)	48.4 (751)	
north-eastern	25.6 (3238)	50.0 (1619)	50.0 (1619)	
north-western	15.9 (2015)	50.5 (1017)	49.5 (998)	
**Physical activity in leisure time** **:**				
low	11.7 (1484)	55.1 (817)	44.9 (667)	<0.0001
moderate	43.8 (5543)	44.4 (2460)	55.6 (3083)	
high	44.4 (5616)	56.0 (3146)	44.0 (2470)	
**Screen time—average per week**				
<2 h/day	33.3 (4209)	47.4 (1996)	52.6 (2213)	<0.0001
2–4 h/day	37.1 (4697)	50.6 (2379)	49.4 (2318)	
>4 h/day	29.6 (3737)	54.8 (2048)	45.2 (1689)	
**Sleep time**—**average per week**				
<6 h/day	10.1 (1276)	47.1 (601)	52.9 (675)	<0.0001
6–8 h/day	50.2 (6342)	48.6 (3082)	51.4 (3260)	
>8 h/day	39.7 (5025)	54.5 (2740)	45.5 (2285)	
**Frequency of eating meals with the family**				
<1 day/week	9.0 (1133)	48.7 (552)	51.3 (581)	0.1165
1–2 days/week	15.0 (1894)	48.6 (920)	51.4 (974)	
3–4 days/week	21.4 (2704)	51.5 (1392)	48.5 (1312)	
5–6 days/ week	16.1 (2038)	52.5 (1070)	47.5 (968)	
every day and more	38.5 (4874)	51.1 (2489)	48.9 (2385)	
**Frequency of consumption of milk and dairy products**				
low	52.8 (6671)	49.7 (3313)	50.3 (3358)	0.0067
high	47.2 (5972)	52.0 (3110)	48.0 (2862)	
**WHtR ^a^**	0.44 (0.43, 0.44)	0.46 (0.45, 0.46)	0.43 (0.42, 0.43)	0.04
**Central obesity ^b^**	16.8 (2125)	18.2 (1169)	13.6 (846)	0.002
**BMI category ^c^**				0.9550
underweight	11.4 (1441)	11.3 (726)	11.5 (715)	
normal	73.3 (9267)	73.3 (4708)	73.2 (4553)	
overweight	15.3 (1934)	15.4 (989)	15.3 (952)	

* chi-square test. ^a^ means (95% confidence interval). ^b^ central obesity identified as waist/height ratio ≥0.5 according to Ashwell et al. [30]. ^c^ body mass index (BMI) categorized according to age/sex-specific cut-offs for Polish children [29]: underweight BMI < 18.5 kg/m^2^; normal weight BMI = 18.5 to 24.9 kg/m^2^; overweight BMI ≥ 25 kg/m^2^.

**Table 2 nutrients-16-02654-t002:** Frequency of the consumption of milk and dairy products according to the opinion of the children and of their parents.

Frequency	Total		*p*-Value	Paired (Children-Parents) N = 6668		*p*-Value *	Consistency of Answers in Family Pairs
	Children N = 12,643 %	Parents N = 7363 %		Children %	Parents %		
Every day and more	29.6	44.8	<0.001	30.1	44.5	<0.001	18.1
Less than every day	68.0	65.0		66.8	64.7		42.9
Never or almost never **	3.4	1.2		3.1	1.1		0.5

* chi-square test; ** including allergy and other factors.

**Table 3 nutrients-16-02654-t003:** Frequency of the consumption of milk and dairy products according to BMI category in the opinion of the children.

BMI Category	Total Group N = 12,643			Boys N = 6423			Girls N = 6220		
	Frequency			Frequency			Frequency		
	Every Day and More	Less than Every Day	Never or Almost Never **	Every Day and More	Less than Every Day	Never or Almost Never **	Every Day and More	Less than Every Day	Never or Almost Never **
underweight	11.6 (438)	11.4 (959)	11.5 (52)	12.3 (243)	11.0 (455)	10.2 (32)	10.8 (194)	11.8 (500)	13.1 (25)
normal	73.2 (2765)	73.2 (6158)	73.4 (333)	71.1 (1404)	74.3 (3075)	73.1 (227)	75.4 (1351)	72.2 (3058)	73.8 (142)
overweight	15.2 (574)	15.4 (1296)	15.1 (69)	16.6 (328)	14.7 (608)	16.7 (52)	13.8 (247)	16.0 (678)	13.1 (25)
***p*-Value ***	0.9994			0.3142			0.2862		

* chi-square test; ** including allergy and other factors.

**Table 4 nutrients-16-02654-t004:** Nutrition knowledge of children and their parents.

Variables	Total		*p*-Value *	Paired (Children-Parents) N = 6668		*p*-Value *	Consistency of Answers in Family Pairs
	Children N = 12,643 %	Parents N = 7363 %		Children %	Parents %		
Nutrition knowledge							
low	27.7	3.4	<0.001	25.9	3.2	<0.001	1.6
medium	64.2	29.1		60.5	28.9		17.1
high	8.1	67.5		13.6	67.9		11.3
Milk and dairy products-answer							
proper	43.6	88.3	<0.001	43.9	88.7	<0.001	39.8
improper	56.4	11.7		56.1	11.3		7.2

* chi-square test.

**Table 5 nutrients-16-02654-t005:** Predictive model for more frequent (at least 5–6 times a week) consumption of milk and dairy products by children.

Parameter	Level	β *	e^β^ **	95% Wald CI		*p*-Value ***
Intercept		−0.953				<0.0001
Nutrition knowledge	high	0.509	1.665	1.44	1.92	<0.0001
	medium	0.198	1.219	1.12	1.32	<0.0001
	low (ref.)	0	1			
Self-assessment of nutrition		0.126	1.135	1.08	1.19	<0.0001
Physical activity in leisure time	high	0.228	1.256	1.11	1.42	0.0002
	moderate	0.083	1.086	0.96	1.22	0.1716
	low (ref.)	0	1			
Sleep time	<6 h/day	0.292	0.747	0.65	0.85	<0.0001
	6–8 h/day	0.214	0.808	0.75	0.87	<0.0001
	>8 h/day (ref.)	0	1			
Gender	girls	−0.072	0.93	0.86	0.99	0.0476
	boys (ref.)	0	1			
Age (years)	10	0.016	1.016	0.93	1.10	0.7201
	11	0.015	1.015	0.93	1.11	0.7318
	12 (ref.)	0	1			
Place of living	<500,000 residents	0.079	1.083	1.04	1.18	0.0386
	>500,000 residents	0.081	1.084	0.96	1.23	0.199
	village (ref.)	0	1			
Frequency of eating meals with the family	1–2 days/week	0.032	1.032	0.83	1.28	0.7748
	3–4 days/week	0.048	1.049	0.84	1.29	0.659
	5–6 days/week	0.155	1.168	0.94	1.45	0.1619
	every day	0.356	1.428	1.16	1.76	0.0008
	<1 day/week (ref.)	0	1			

* model parameter (β); ** OR—point estimate (e^β^), 95% confidence intervals; *** significance level of Wald’s test.

**Table 6 nutrients-16-02654-t006:** Prediction model for more frequent (at least 5–6 times a week) consumption of milk and dairy products by children based on parents’ opinions.

Parameter	Level	β *	e^β^ **	95% Wald CI		*p*-Value ***
Intercept		−0.945				0.0003
Nutrition knowledge	high	0.694	2.002	1.52	2.64	<0.0001
	medium	0.397	1.487	1.12	1.97	0.0036
	low (ref.)	0	1			
Self-assessment of nutrition		0.183	1.206	1.12	1.29	<0.0001
Physical activity in leisure time	high	0.126	1.134	0.96	1.35	0.1498
	moderate	0.049	1.050	0.89	1.24	0.5632
	low (ref.)	0	1			
Sleep time	<6 h/day	−0.676	0.509	0.27	0.95	0.0339
	6–8 h/day	−0.266	0.766	0.69	0.85	<0.0001
	>8 h/day (ref.)	0	1			
Gender (sex)	girls	−0.078	0.925	0.84	0.98	0.0423
	boys (ref.)	0	1			
Age (years)	10	−0.23	0.795	0.71	0.89	0.0001
	11	−0.155	0.856	0.76	0.97	0.0118
	12 (ref.)	0	1			
Place of living	<500,000 residents	0.066	1.068	0.95	1.2	0.2726
	>500,000 residents	0.324	1.383	1.16	1.65	0.0004
	village (ref.)	0	1			
Frequency of eating meals with the family	1–2 days/week	0.185	1.203	0.85	1.70	0.2937
	3–4 days/week	0.244	1.276	1.09	1.78	0.0336
	5–6 days/week	0.357	1.428	1.02	2.01	0.04
	every day	0.429	1.536	1.10	2.14	0.0111
	<1 day a week (ref.)	0	1			

* model parameter (β); ** OR—point estimate (e^β^), 95% confidence intervals; *** significance level of Wald’s test.

## Data Availability

The data presented in this study are available upon request from the corresponding author.

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
