# Peer review of "What Determinants Are Related to Milk and Dairy Product Consumption Frequency among Children Aged 10–12 Years in Poland? Nationwide Cross-Sectional Study"

_nutrients, 2024, doi:10.3390/nu16162654_

Round 1

Reviewer 1 Report (Previous Reviewer 1)

Comments and Suggestions for Authors

Although the manuscript provided has some strengths, like large sample size (over 12,000 children and 7,000 parents) that provides good statistical power and representativeness, there seems to be fundamental limitations of the research:

1. Cross-sectional design limits ability to determine causality between factors and dairy consumption.

2. Reliance on self-reported data for dietary intake and other measures, which is subject to recall bias and social desirability bias.

3. Use of frequency questionnaires rather than more detailed dietary assessment methods (e.g. 24-hour recalls) limits precision in quantifying actual dairy intake amounts.

4. Lack of consideration of family economic status as a potential confounding factor.

5. No assessment of reasons for dairy avoidance (e.g. lactose intolerance, milk allergy) or use of dairy alternatives.

6. Limited generalizability outside of Poland due to potential cultural differences in dairy consumption patterns.

7. Discrepancies between parent and child reports of dairy consumption frequency highlight potential reliability issues.

8. Categorization of consumption frequency into broad groups (e.g. "low" vs "high") may obscure more nuanced patterns.

9. No data on actual nutritional status or health outcomes to correlate with dairy intake.

10. Nutrition knowledge assessment based on limited number of questions may not fully capture nutrition literacy.

While this study provides valuable insights into factors associated with dairy consumption among Polish adolescents, with nationwide study covering all regions of Poland which increases generalizability, the issues seems to be solved before the decision.

Author Response

Dear Reviewer,

Thank you for your diligent work on our manuscript entitled “Determinants of the Frequency of Milk and Dairy Products Consumption by Adolescents aged 10-12 in Poland in the Opinions of Children and their Parents in Relation to the Level of their Nutrition Knowledge. Nationwide Cross-sectional Study. Your comments have significantly improved our paper and provided valuable suggestions for future research. Please find our responses to your comments below.

Authors

Comments Overall:

Although the manuscript provided has some strengths, like a large sample size (over 12,000 children and 7,000 parents) that provides good statistical power and representativeness, there seem to be fundamental limitations of the research:

Response:

Thank you very much for your comments which helped improve our manuscript's quality. We hope you will find our responses relevant to your comments. We made every effort to address all the suggestions accordingly, added more details, and improved our paper.

Comments 1:

  1. Cross-sectional design limits ability to determine causality between factors and dairy consumption.

Response 1:

Thank you for your comment. Ultimately, we share your point of view and agree that cross-sectional design limits the possibility of establishing a causal relationship between the factors studied and the outcomes including food consumption, as well as milk and dairy consumption.

However, this type of study is often used. It has advantages and disadvantages, which should be considered when planning research and interpreting the results.

We are aware of the limitations of our study. Still, we should consider that these are nationwide studies with a stratified sampling design (a two-stage cluster sampling design was used to select a representative sample of students aged 10-12 years from public and private schools. The first stage involved selecting schools with a “probability proportional to macroregion and the size of the place,” while the second stage involved randomly selecting classes within each school. In the selected classes, all students were eligible to participate in the survey; however, not all parents agreed to their children's participation in the research. This is a valuable study for this age group and the population of children in Poland.

 In addition, this was emphasized in the limitations, citing, among others, the publication by Wang et al., 2020 (Wang X., Cheng Z., Cross-Sectional Studies. Strengths, Weaknesses, and Recommendations. Chest Journal 2020; 158(1S):S65-S71, DOI: https://doi.org/10.1016/j.chest.2020.03.012.)

Comments 2:

Reliance on self-reported data for dietary intake and other measures, which is subject to recall bias and social desirability bias.

Response 2:

 Of course, this is a weakness of this type of study because gathering information about behaviors is only useful if participants are willing to disclose real nutrition behavior to the researcher. Participants may try to give the ‘correct’ responses they think researchers are looking for (or deliberately do the opposite) or try to come across in the most socially acceptable way (i.e., social desirability bias), which can lead to giving untruthful responses.

It should be emphasized, however, that this is a common method of collecting data, and the study was conducted in accordance with generally accepted procedures (Cade et al., 2017; Dao et al., 2019; FAO, 2018; Golley et al., 2017)

  • Cade, J.E., Warthon-Medina, M., Albar, S. et al. DIET@NET: Best Practice Guidelines for dietary assessment in health research. BMC Med 15, 202 (2017). https://doi.org/10.1186/s12916-017-0962-x
  • Dao MC, Subar AF, Warthon-Medina M, Cade JE, Burrows T, Golley RK, Forouhi NG, Pearce M, Holmes BA. Dietary assessment toolkits: an overview. Public Health Nutr. 2019 Mar;22(3):404-418. doi: 10.1017/S1368980018002951.
  • Food and Agriculture Organization. Dietary Assessment: A Resource Guide to Method Selection and Application in Low Resource Settings. 2018. Available online: http://www.fao.org/3/i9940en/I9940EN
  • Golley RK, Bell LK, Hendrie GA, Rangan AM, Spence A, McNaughton SA, et al. (2017): Validity of short food questionnaire items to measure intake in children and adolescents: a systematic review. J Hum Nutr Diet, 30(1): 36–50.

Comments 3:

Use of frequency questionnaires rather than more detailed dietary assessment methods (e.g. 24-hour recalls) limits precision in quantifying actual dairy intake amounts.

Response 3:

We are very grateful for this suggestion and agree it would be a good option. However, as it would be difficult to use a quantitative method to collect data on consumption in such a large population (over 12,000 children and over 7,000 parents), we decided to use the FFQ questionnaire, commonly used in this type of research.

The most accurate option would be to conduct a 3-4 day dietary recall, which would reflect the actual consumption of food products and nutrients, but this would be much more time-consuming and expensive. We are planning such a study in a smaller group, and we do not know if it will be representative.

Comments 4:

  1. Lack of consideration of family economic status as a potential confounding factor.

Response 4:

Thank you for your suggestion.

In our analysis, income in the case of milk and dairy products was not taken into account because children in the Polish countryside usually have greater access to these products from their own farms than children from cities, despite lower household incomes. In turn, in cities, it has been fashionable for many years to consume lactose-free products and avoid milk and dairy products, despite higher family incomes. Based on the research analyses and the literature review [Wiza, 2022], it was shown that changes in the level of milk consumption were small in relation to changes in prices expressed in current prices, as well as consumer income. In contrast, there were changes in dairy consumption. Between 2004 and 2020, the consumption of highly processed dairy products, i.e., yogurts, cheeses, and cottage cheese, increased. Meanwhile, the consumption of liquid milk decreased, and the consumption of fat products, i.e., butter and cream, increased at a lower rate. Therefore, analyzing the data in this respect would not be justified.

Wiza P., Consumption of milk and milk products in Poland in the years 2004-2020. Technological Progress in food processing (Postępy technolo przetwórstwa spożywczego), 2022, 2, 5-10.

All other studied socio-demographic factors were used in statistical analyzes and presented in this manuscript.

Family economic status as a potential confounding and limiting factor was presented in the limitations and add

Comments 5:

No assessment of reasons for dairy avoidance (e.g., lactose intolerance, milk allergy) or use of dairy alternatives.

Response 5:

The study did not consider factors related to the elimination of milk and dairy products due to lactose intolerance or milk protein allergy, but our data showed that only 1-3% of respondents did not consume milk and dairy products. We have included this in the limitations as an example to be included in other future studies.

Thank you very much for this comment and suggestion. It is very valuable for this type of study.

Comments 6:

Limited generalizability outside of Poland due to potential cultural differences in dairy consumption patterns.

Response 6:

Thank you for your suggestion. We added another limitation as follows:

The limitation is the lack of generalizability outside of Poland due to potential cultural differences in dairy consumption patterns.

Comments 7:

  1. Discrepancies between parent and child reports of dairy consumption frequency highlight potential reliability issues.

Response 7:

Thank you for this comment and for pointing this out. “Future research should use mixed methods that include both self-report and direct observations to minimize discrepancies between parents' and children's perceptions”. However, this may be difficult in very large cohort studies. However, we would like to point out that the discrepancies between parents and children result from the fact that children see the world differently than their parents. Children completed surveys at school and in the classroom, while parents often answered about the actual consumption of milk and dairy products by parents. According to the authors, separating the research into parents and children allowed us to avoid influencing the responses of one group (children) to the other (parents). Neither group could agree on the answer among themselves. This result could be disturbed if they completed the questionnaire together by prompting the children to answer the questions by their parents, and this is how we know the actual state of knowledge of the children and their parents, as well as the frequency of consumption of these products by the children. In our opinion, these differences cannot be eliminated. This research method facilitates (realizes) the interpretation of the obtained results and allows for the inclusion of further educational activities aimed at both children and their parents.

Comments 8:

Categorization of consumption frequency into broad groups (e.g. "low" vs "high") may obscure more nuanced patterns.

Response 8:

Thank you for this comment and for pointing this out.

We conducted detailed analyses and found no statistically significant differences, which is why we decided to present these results in 2 categories low and high frequency of milk and dairy product consumption. We decided to present the results in this way (analyze these results) because the recommendations refer to daily consumption. In addition, we wanted to present/highlight the low frequency of milk consumption every day and

Adolescence, a critical period marked by significant physical and cognitive development, also sees the establishment of dietary preferences and habits that are likely to persist into adulthood. Consequently, the nutritional decisions made during this formative phase can set the stage for an individual’s health trajectory well into the future.  more in the group of school children in the context of their impact on future health.

Comments 9:

No data on actual nutritional status or health outcomes to correlate with dairy intake.

Response 9:

Thank you for this comment.

We have performed additional analyses and included nutritional status: BMI-for-age was categorized according to Polish sex-specific cut-offs for children [Kułaga et al., 2015]. BMI-for-age ≥25 kg/m2 was used as an overweight/obesity measure (general adiposity marker), and WHtR ≥0.5 as a central obesity measure (central adiposity marker) [Ashwell et al., 2012].

We have added the relevant data in Material and Methods and in Results - table 1. In addition, we have added Table 3. Frequency of consumption of milk and dairy products according to BMI Category in the opinion of the children.

However, given that these indicators did not significantly affect the dependent variables in the models, they were not included/presented in Tables 5 and 6. This is reported in Section 2.5 Statistical Analysis, lines 236-240.

We hope that our manuscript is now more complete and addresses the suggestions made by the reviewer.

Kułaga, Z.; Różdżyńska-Świątkowska, A.; Grajda, A.; Gurzkowska, B.; Wojtyło, M.; Góźdź, M.; Świąder-Leśniak, A.; Litwin, M. Percentile charts for growth and nutritional status assessment in Polish children and adolescents from birth to 18 year of age. Stand. Med. Pediatr. 2015, 12, 119–135.

Ashwell, M.; Gunn, P.; Gibson, S. Waist-to-height ratio is a better screening tool than waist circumference and BMI for adult cardiometabolic risk factors: Systematic review and meta-analysis. Obes. Rev. 2012, 13, 275–286. doi: 10.1111/j.1467-789X.2011.00952.x.

Comments 10:

Nutrition knowledge assessment based on limited number of questions may not fully capture nutrition literacy.

Response 10:

Our study used 20 questions regarding nutritional knowledge, which we consider sufficient. The questions were tested in pilot studies and previous studies, and this was described in detail in the Protocol study (Hamulka et al. 2024 – ref.22). In addition, an excess of questions often causes respondent fatigue, which does not affect the quality and accuracy of the obtained results. Nutritional knowledge was not the main goal of this study but only a supplement.

While this study provides valuable insights into factors associated with dairy consumption among Polish adolescents, with a nationwide study covering all regions of Poland, which increases generalizability, the issues seem to be solved before the decision.

Reviewer 2 Report (New Reviewer)

Comments and Suggestions for Authors

The study carried out is very interesting and provides very important information regarding dairy consumption in children (10 - 12 years). The methodology used is sufficient and allows the objective of the study to be evaluated. The results support the discussion. However, I have the following comments.

I. Comments.

1. Are children between 10 and 12 years old children or adolescents?

2. Improve the wording of the objective of the study.

3. Various studies have demonstrated the relevance of dairy products to promote better nutrition (prevent malnutrition, overweight or obesity) in children and adolescents. It is important that the authors include this point briefly in the introduction.

4. A portion of dairy products provides various nutrients, and if two or three are consumed, it is more feasible to reach the nutrient recommendation in a child or adolescent. This point should be discussed by the authors.

5. Would it be possible for the authors to include the nutrient contribution, according to the type of dairy consumed? It would be ideal if they could compare the dairy intake with the nutrient contribution per day.

6. Improve the resolution of Figure 1.

7. Some studies have associated dairy consumption with a better quality of diets and better lifestyles in children and adolescents (less alcohol consumption or sedentary lifestyle). I suggest discussing this point.

8. What projections would this study have, especially in education, nutrition and health?

Comments on the Quality of English Language

Review the wording of the manuscript, and correct some errors in the writing.

Author Response

Dear Reviewer,

Thank you for your diligent work on our manuscript entitled “Determinants and Nutritional Implications Related to the Frequency of Milk and Dairy Products Consumption by Adolescents aged 10-12 in Poland. Nationwide Cross-sectional Junior-Edu-Żywienie (JEŻ) study. Your comments have significantly improved our paper and provided valuable suggestions for future research. Please find our responses to your comments below.

Authors

Comments Overall:

The study carried out is very interesting and provides very important information regarding dairy consumption in children (10 - 12 years). The methodology used is sufficient and allows the study's objective to be evaluated. The results support the discussion. However, I have the following comments.

Comments 1:

Are children between 10 and 12 years old children or adolescents?

Response 1:

Thank you, we concur with your suggestion. We have included appropriate changes throughout the manuscript, using the term 'children aged 10-12' or 'schoolchildren'.

Comments 2:

Improve the wording of the objective of the study.

Response 2:

Thank you for your suggestion. We have revised the aim according to your recommendation:

The study aimed to identify (i) the frequency of milk and dairy products consumption by children aged 10-12 and (ii) factors related to the consumption of these products, including their lifestyle, nutrition knowledge, and nutritional status, as well as parents' nutrition knowledge.

Comments 3:

Various studies have demonstrated the relevance of dairy products to promote better nutrition (prevent malnutrition, overweight or obesity) in children and adolescents. It is important that the authors include this point briefly in the introduction.

Response 3:

Thank you for this comment.

We have performed additional analyses and included nutritional status: BMI-for-age was categorized according to Polish sex-specific cut-offs for children [Kułaga et al., 2015]. BMI-for-age ≥25 kg/m2 was used as an overweight/obesity measure (general adiposity marker), and WHtR ≥0.5 as a central obesity measure (central adiposity marker) [Ashwell et al., 2012].

We have added the relevant data in Material and Methods, and in Results - table 1. In addition, we have added Table 3. Frequency of consumption of milk and dairy products according to BMI Category in the opinion of the children.

However, given that these indicators did not significantly affect the dependent variables in the models, they were not included/presented in tables 5 and 6. This is reported in Section 2.5 Statistical Analysis, lines 236-240.

There was also no significant correlation between milk and milk product consumption and obesity indices (BMI - general adiposity marker; WHtR -central adiposity marker), so we did not discuss this issue extensively.

In the introduction, it was described that study results clearly indicate that milk and yogurt consumption was associated with healthy growth in adolescents' children height and weight without increasing the risk of obesity [1, 13-15].

We hope that our manuscript is now more complete and addresses the suggestions made by the reviewer.

Comments 4:

 A portion of dairy products provides various nutrients, and if two or three are consumed, it is more feasible to reach the nutrient recommendation in a child or adolescent. This point should be discussed by the authors.

Response 4:

Thank you for pointing this out. We expanded discussion as you suggested.

Comments 5:

Would it be possible for the authors to include the nutrient contribution according to the type of dairy consumed? It would be ideal if they could compare the dairy intake with the nutrient contribution per day.

Response 5:

We are very grateful for this suggestion and agree that this would be a very good option. However, from the data obtained in this study, this is not possible because we used only the FFQ questionnaire to collect data on food consumption. In addition, in such a large study group (over 12,000 children), it would be difficult to conduct a 3-4 day dietary recall (only this would reflect the actual consumption of these products and nutrients). Even the use of a 24-hour interview would not be reliable and fully characterise the consumption of foods and nutrients in the group of children and adolescents.

Comments 6:

Improve the resolution of Figure 1.

Response 6:

Thank you. We changed it.

Comments 7:

Some studies have associated dairy consumption with a better quality of diets and better lifestyles in children and adolescents (less alcohol consumption or sedentary lifestyle). I suggest discussing this point.

Response 7:

Thank you for your suggestion.

We included selected lifestyle elements, such as physical activity in leisure time, screen time, sleep time, and frequency of eating meals with the family, and discussed this in both the results and the discussion

As the research was conducted at school in the presence of teachers, we did not ask about the use of stimulants, including alcohol consumption - this would not be reliable data. We agree that it would be interesting, but perhaps other studies should focus more on the effect of stimulant use on food consumption, including milk and dairy products.

Comments 8:

What projections would this study have, especially in education, nutrition, and health?

Response 8:

Thank you for this question. Our study is very important in the context of nutrition education in schools on healthy lifestyles, including increasing consumption of milk and dairy products as important sources of calcium, protein, and vitamins.

Therefore, it was highlighted in the section: Conclusions and Implications (lines 571-599).

Round 2

Reviewer 1 Report (Previous Reviewer 1)

Comments and Suggestions for Authors

All comments were addressed.

Reviewer 2 Report (New Reviewer)

Comments and Suggestions for Authors

The authors responded to all my questions and comments.

This manuscript is a resubmission of an earlier submission. The following is a list of the peer review reports and author responses from that submission.

Round 1

Reviewer 1 Report

Comments and Suggestions for Authors

Dear authors,

I have now completed the review of the manuscript titled "Determinants and Nutritional Implications Related to the Frequency of Milk and Dairy Products Consumption by Adolescents aged 10-12 in Poland. Nationwide Cross-sectional Junior-Edu -Żywienie (JEŻ) study."

In the present study, you provide valuable insights into factors associated with milk and dairy intake in Polish adolescents using a large representative sample. The large sample size of over 12,000 participants covering the entire territory of Poland allows for generalization to the broader adolescent population aged 10-12. The study also accounted for important sociodemographic factors, nutritional knowledge of students and parents, and lifestyle factors in the statistical models. Collecting data from both children and parents enabled determining compliance in reporting milk and dairy consumption.

The manuscript is interesting and, in general, fairly well-written. However, I have some suggestions to further improve the quality of the manuscript. I would like to suggest that you address these limitations in the article, either by discussing them in the limitations section or, where feasible, by making the appropriate revisions:

1. Physical activity among adolescents was identified as one factor influencing dairy consumption in the study. However, trends in physical activity among adolescents in 2009-2021 are very different. Can you find the trend changes and discuss whether your idea is still valid in 2024?

2. Breastfeeding and complementary foods may be related to the factors influencing milk/dairy intake. Can you show me data related to this? Or you can discuss these ideas at least.

3. Some readers may suggest that using convenience sampling rather than random sampling will introduce potential selection bias that could impact generalizability. What is your opinion on this?

Thank you for your valuable contributions to our field of research. I look forward to receiving the revised manuscript.

Author Response

Response to Reviewer 1

Dear Reviewer,

Thank you for your diligent work on our manuscript entitled “Determinants and Nutritional Implications Related to the Frequency of Milk and Dairy Products Consumption by Adolescents aged 10-12 in Poland. Nationwide Cross-sectional Junior-Edu-Żywienie (JEŻ) study. Your comments have significantly improved our paper and provided valuable suggestions for our future research. Please find our responses to your comments below.

Comments Overall:

I have now completed the review of the manuscript titled "Determinants and Nutritional Implications Related to the Frequency of Milk and Dairy Products Consumption by Adolescents aged 10-12 in Poland. Nationwide Cross-sectional Junior-Edu -Żywienie (JEŻ) study."

 In the present study, you provide valuable insights into factors associated with milk and dairy intake in Polish adolescents using a large representative sample. The large sample size of over 12,000 participants covering the entire territory of Poland allows for generalization to the broader adolescent population aged 10-12. The study also accounted for important sociodemographic factors, nutritional knowledge of students and parents, and lifestyle factors in the statistical models. Collecting data from both children and parents enabled determining compliance in reporting milk and dairy consumption.

The manuscript is interesting and, in general, fairly well-written. However, I have some suggestions to further improve the quality of the manuscript. I would like to suggest that you address these limitations in the article, either by discussing them in the limitations section or, where feasible, by making the appropriate revisions:

Response:

Thank you very much for your comments which helped to improve the quality of our manuscript. We hope you will find our responses relevant to your comments. We made every effort to address all the suggestions accordingly, added more details, and improved our paper.

Comments 1:

Physical activity among adolescents was identified as one factor influencing dairy consumption in the study. However, trends in physical activity among adolescents in 2009-2021 are very different. Can you find the trend changes and discuss whether your idea is still valid in 2024?

Response 1:

Thank you for this comment.

Research shows that there is a connection between healthy dietary patterns, including the consumption of milk and dairy products, and physical activity - in our article, approximately 90% had moderate and high activity.

In addition, higher physical activity combined with higher milk consumption affects bone health in this population group. Available data indicated that childhood  and youth  (2–19 years) dairy product consumption may affect various facets of growth and development, and scientific evidence suggests that dairy products are important for linear growth and bone health [Dror and Allen, 2014; Lamas et al., 2019].

Furthermore, in Spain study indicated, the association between dairy consumption and dietary patterns and intake of selected nutrients in 7–11-year-old children. Children who drank more milk also had better dietary patterns [Ortega et al., 2012]. Similar trends were observed in the group of younger children aged 1-10 years (the EsNuPI study) [Madrigal et al., 2019].

Moreover, was conducted the relationship between specific lifestyle behaviors and dairy consumption in a sample of European children and adolescents from the IDEFICS study. The study found that European children with a healthy lifestyle, specifically regarding physical activity and sedentary behaviors over time, consumed more milk and yogurt. This study suggests that the protective effect of dairy products on cardiovascular diseases (CVD) and other  non-communicable chronic diseases could be related to the association between their consumption and specific lifestyle behaviors [Santaliestra-Pasías eta al., 2019].

  • Dror, D.K.; Allen, L.H. Dairy product intake in children and adolescents in developed countries: Trends, nutritional contribution, and a review of association with health outcomes. Nutr. Rev. 2014, 72, 68–81. https://doi.org/10.1111/nure.12078
  • Lamas, C.; Castro, M.J.; Gil-Campos, M.; Gil, A.; Couce, M.L.; Leis, R. Effects of dairy product consumption on height and bone mineral content in children: A systematic review of controlled trials. Adv. Nutr. 2019, 10, 88–96. https://doi.org/10.1093/advances/nmy096
  • Hamrani A, Mehdad S, El Kari K, El Hamdouchi A, El Menchawy I, Belghiti H, El Mzibri M, Musaiger AO, Al-Hazzaa HM, Hills AP, Mokhtar N, Aguenaou H. Physical activity and dietary habits among Moroccan adolescents. Public Health Nutr. 2015 Jul;18(10):1793-800. doi: 10.1017/S1368980014002274
  • Ortega, R.O.; González-Rodríguez, L.; Jiménez Ortega, A.; Perea Sánchez, J.; Bermejo López, L. N°920030, G. de investigación. Implicación del consumo de lácteos en la adecuación de la dieta y de la ingesta de calcio y nutrientes en niños españoles. Nut. Clin. Diet Hosp. 2012, 32, 28–36. [Google Scholar]
  • Madrigal, C.; Soto-Méndez, M.J.; Hernández-Ruiz, Á.; Ruiz, E.; Valero, T.; Ávila, J.M.; Lara-Villoslada, F.; Leis, R.; Martínez de Victoria, E.; Moreno, J.M.; et al. Dietary and Lifestyle Patterns in the Spanish Pediatric Population (One to <10 Years Old): Design, Protocol, and Methodology of the EsNuPI Study. Nutrients 2019, 11, 3050. https://doi.org/10.3390/nu11123050
  • Santaliestra-Pasías, A.M.; González-Gil, E.; Pala, V.; Intemann, T.; Hebestreit, A.; Russo, P.; Van Aart, C.; Rise, P.; Veidebaum, T.; Molnar, D.; et al. Predictive associations between lifestyle behaviours and dairy consumption: The IDEFICS study. Nutr. Metab. Cardiovasc. Dis. 2019, 19, 1–24. [Google Scholar]

On the other hand in published this year in a systematic literature [Ziemba et al. 2024] the available data related to physical activity (PA) and sedentary behaviour (SB) in Polish children and adolescents have confirmed the non-fulfilment of PA recommendations for maintaining health and reducing sedentary behaviour. The authors draw particular attention to the importance of physical education in meeting PA recommendations. Failure to meet PA recommendations directly translates into reduced health potential and increases the likelihood of overweight and obesity. It is worth noting that the effectiveness of PA and SB research in children and adolescents requires the involvement of the entire educational environment. The importance of the school, especially physical education, and the family as an educational environment in PA efforts is invaluable. Structured top-down government action is needed. Achieving PA recommendations should be prioritised and set against sedentary behaviour.

Ziemba, M., Groffik, D., Frömel, K.,  & Vorlíček, M. (2024). SURVEILLANCE OF PHYSICAL ACTIVITY AND SEDENTARY BEHAVIOR IN POLISH CHILDREN AND ADOLESCENTS: A SCOPING REVIEW OF LITERATURE FROM 2000 TO 2022. Health Problems of Civilization, 18(2), 215-244. https://doi.org/10.5114/hpc.2023.133886

Despite changing trends in physical activity (PA) among primary school students in Europe, including Poland, there is still evidence that PA is too low, especially after the pandemic, and that students still spend too much time in front of screens [Zembura et al. 2022].

  1. Zembura, P.; Korcz, A.; Nałęcz, H.; Cieśla, E. Results from Poland’s 2022 Report Card on Physical Activity for Children and Youth. Int. J. Environ. Res. Public Health 2022, 19, 4276. https:// doi.org/10.3390/ijerph19074276.

Comments 2:

Breastfeeding and complementary foods may be related to the factors influencing milk/dairy intake. Can you show me data related to this? Or you can discuss these ideas at least.

Response 2:

Thank you for your point of view.

Breastfeeding and complementary foods as factors influencing the consumption of milk and products by adolescents were not the subject of our considerations/research. Therefore, we cannot show you such data and statistical analyzes taking this data into account.

In the scientific literature, the impact/relationship between breastfeeding and/or complementary foods is analyzed, but mainly in the younger age group - up to 2-3 years of age or in a group of older children (usually preschool and early school children) [Hong et al., 2023; Kostecka et al., 2021] or in relation to dietary intake. fruits and vegetables. It should be noted, however, that breastfeeding was much more likely to have a positive effect on the consumption of vegetables and, rather, fruit [Hamulka et al., 2020; Mennella et al., 2019].

  • Hong J, Chang JY, Oh S, Kwon SO. Growth, Nutritional Status, and Dietary Intake Patterns Associated With Prolonged Breastfeeding in Young Korean Children: A Nationwide Cross-Sectional Study. J Korean Med Sci. 2023 Apr 17;38(15):e116. doi: 10.3346/jkms.2023.38.e116.
  • Kostecka M, Jackowska I, Kostecka J. A Comparison of the Effects of Young-Child Formulas and Cow's Milk on Nutrient Intakes in Polish Children Aged 13-24 Months. Nutrients. 20
  • Hamulka, J.; Zielinska, M.A.; Jeruszka-Bielak, M.; Górnicka, M.; Głąbska, D.; Guzek, D.; Hoffmann, M.; Gutkowska, K. Analysis of Association between Breastfeeding and Vegetable or Fruit Intake in Later Childhood in a Population-Based Observational Study. Int. J. Environ. Res. Public Health 2020, 17, 3755. https://doi.org/10.3390/ijerph17113755
  • Mennella, J.A.; Daniels, L.M.; Reiter, A.R. Learning to like vegetables during breastfeeding: A randomized clinical trial of lactating mothers and infants. Am. J. Clin. Nutr. 2017, 106, 67–76.

Comments 3:

Some readers may suggest that using convenience sampling rather than random sampling will introduce potential selection bias that could impact generalizability. What is your opinion on this?

Response 3:

We agree with your opinion. We agree that random sampling would be best, allowing full generalization of the data and appropriate conclusions.

We used stratified sampling - macroregion/school/class and age of children, which, given the large number of students surveyed, makes the sample representative and makes it possible to generalize the conclusions to the entire population of young people in Poland in this age group (10-12 years old). We sent an inquiry to all primary schools (n=14322) in Poland, but 2,208 schools responded positively, of which parents in not all schools agreed to their children's participation in the research.

We have added an appropriate explanation in the text, in the section 2.2. Study Design and Participants

Reviewer 2 Report

Comments and Suggestions for Authors

The article is robust, relevant, and well-founded. It is suggested to improve the limitations section by emphasizing the cross-sectional nature of the study and the possible differences in the perception of parents and children.

1.       Clarify the cross-sectional nature of the study and how this might influence the interpretation of the results.

2.       Address the discrepancy between parents' and children's perceptions, offering possible explanations or proposing methods for future research to minimize this variability.

3.       Improve the documentation of the questionnaire used, providing examples of questions and explaining how it was validated to ensure its reliability and validity.

Limitations of the study:

1.       Cross-sectional design: This type of study limits the ability to establish causality between the factors studied and dairy consumption.

2.       Self-reported questionnaires: Information based on self-reporting may be subject to recall bias and social desirability bias.

3.       Disagreement between parents and children: There is a significant discrepancy between parents' and children's perceptions of consumption frequency, which could affect the interpretation of the results.

Recommendations for Improving the Article

1. Clarity on the Nature of the Study:

  • Cross-Sectionality: Ensure that readers understand that the cross-sectional design limits the ability to establish causality. Include a more detailed discussion on how this limitation affects the interpretation of the results.
  • Scope Explanation: Clarify in the introduction and discussion that the results are correlational and cannot infer direct causal relationships between the studied factors and dairy consumption.

2. Improved Documentation of the Questionnaire:

  • Questionnaire Details: Include specific examples of questionnaire questions and explain how it was validated. This could help other researchers replicate the study and verify the reliability and validity of the instrument.
  • Validation Methodology: Provide more details on the validation process of the questionnaire used to ensure that the responses are representative and accurate.

3. Addressing the Discrepancy between Parents' and Children's Perceptions:

  • Comparative Analysis: Conduct a deeper analysis of why there is a significant discrepancy between parents' and children's perceptions of the frequency of dairy consumption.
  • Proposed Solutions: Suggest methods for future research that could minimize this discrepancy, such as using mixed methods that include both self-reporting and direct observations.

4. Inclusion of Longitudinal Analyses:

  • Recommendation for Future Studies: Suggest conducting longitudinal studies that can follow the same subjects over time to establish more robust causal relationships between the studied factors and dairy consumption.
  • Exploration of Temporal Trends: Discuss the possibility of long-term follow-up of the current sample to observe how consumption patterns and associated factors change over time.

5. Expansion of International Context:

  • International Comparisons: Include comparisons with similar studies conducted in other countries to place the results in a global context. This could help identify specific cultural factors influencing consumption patterns.
  • Global Relevance: Discuss how the findings could be applicable or relevant to populations outside of Poland.

6. Deepening the Analysis of Socioeconomic Factors:

  • Socioeconomic Breakdown: Conduct a more detailed analysis of how different socioeconomic levels influence dairy consumption, as this factor can be crucial for the implementation of effective public policies.
  • Specific Interventions: Propose specific interventions aimed at various socioeconomic groups to improve dairy consumption equitably.

7. Improvement in Presentation of Results:

  • Data Visualization: Use more visual and less text-dense charts and tables to present the results more clearly and comprehensibly.
  • Summary of Key Findings: Include a visual summary or infographic highlighting the study’s key findings to facilitate quick understanding of the most important points.

8. Exploration of Family Habits:

  • Impact of Family Meals: Investigate more deeply how the frequency and quality of family meals affect the consumption of dairy products and other healthy foods.
  • Educational Interventions: Propose educational programs involving the whole family to promote healthy eating habits from an early age.

Author Response

Response to Reviewer 2

Dear Reviewer,

Thank you for your diligent work on our manuscript entitled “Determinants and Nutritional Implications Related to the Frequency of Milk and Dairy Products Consumption by Adolescents aged 10-12 in Poland. Nationwide Cross-sectional Junior-Edu-Żywienie (JEŻ) study. Your comments have significantly improved our paper and provided valuable suggestions for our future research. Please find our responses to your comments below.

Comments Overall:

The article is robust, relevant, and well-founded. It is suggested to improve the limitations section by emphasizing the cross-sectional nature of the study and the possible differences in the perception of parents and children.

  1. Clarify the cross-sectional nature of the study and how this might influence the interpretation of the results.
  2. Address the discrepancy between parents' and children's perceptions, offering possible explanations or proposing methods for future research to minimize this variability.
  3. Improve the documentation of the questionnaire used, providing examples of questions and explaining how it was validated to ensure its reliability and validity.

Response:

Thank you for your opinion and comments. We have made every effort to ensure the highest quality of our manuscript. We agree that the suggested changes will contribute to the improvement of our paper. 
As suggested, we included more information into the limitations section.

We have addressed the reviewer's specific issues and suggestions below.

Limitations of the study:

  1. Cross-sectional design: This type of study limits the ability to establish causality between the factors studied and dairy consumption.
  2. Self-reported questionnaires: Information based on self-reporting may be subject to recall bias and social desirability bias.
  3. Disagreement between parents and children: There is a significant discrepancy between parents' and children's perceptions of consumption frequency, which could affect the interpretation of the results.

Response:

Thank you for your point of view. I agree with your suggestions. We have tried to include them in the revised article.

You can find detailed answers in the ‘Recommendations for Improving the Article' section below.

Comments 1:

  1. Clarity on the Nature of the Study:
  • Cross-Sectionality: Ensure that readers understand that the cross-sectional design limits the ability to establish causality. Include a more detailed discussion on how this limitation affects the interpretation of the results.
  • Scope Explanation: Clarify in the introduction and discussion that the results are correlational and cannot infer direct causal relationships between the studied factors and dairy consumption.

Response 1:

Thank you for this suggestion. We share your point of view and agree that Cross-sectional design: This type of study limits the ability to establish causality between the factors studied and dairy consumption.

Consistent with scientific evidence and literature data:

"A cross-sectional study is a type of research design in which you collect data from many different individuals at a single point in time. In cross-sectional research, you observe variables without influencing them. A cross-sectional study is a cheap and easy way to gather initial data and identify correlations that can then be investigated further in a longitudinal study".

This type of research has many advantages and is often used:

  • Because you only collect data at a single point in time, cross-sectional studies are relatively cheap and less time-consuming than other types of research.
  • Cross-sectional studies allow you to collect data from a large pool of subjects and compare differences between groups.
  • Cross-sectional studies capture a specific moment in time.

In turn, the disadvantages include:

  • It is difficult to establish cause-and-effect relationships using cross-sectional studies, since they only represent a one-time measurement of both the alleged cause and effect.
  • Since cross-sectional studies only study a single moment in time, they cannot be used to analyze behavior over a period of time or establish long-term trends.
  • The timing of the cross-sectional snapshot may be unrepresentative of behavior of the group as a whole.

Therefore, this should be taken into account when planning research and interpreting the results obtained.

(Wang X., Cheng Z., Cross-Sectional Studies. Strengths, Weaknesses, and Recommendations. Chest Journal 2020; 158(1S):S65-S71, DOI: https://doi.org/10.1016/j.chest.2020.03.012.)

 We have added an appropriate explanation in the text, in the section 2.2. Study Design and Participants.

In the Project created a representative sample including 1st–6th grade schoolchildren (aged 7–12 years) from schools located in all 16 voivodeships across Poland. The school class was the smallest sampling unit. This approach resulted from the theory that the students underwent the same school education and were at a similar stage of growth. The schools were selected based on the distribution of schools into regions, considering three sizes of the locality (i.e., villages, towns with up to 100,000 inhabitants, and large urban agglomerations). Taking the above into account, five separate macroregions were included: Central (Masovian Voivodeship, Łód´z Voivodeship); North-Eastern (Warmian-Masurian Voivodeship, Podlaskie Voivodeship, Lublin Voivodeship); NorthWest (Pomeranian Voivodeship, West Pomeranian Voivodeship, Kuyavian-Pomeranian Voivodeship and Greater Poland Voivodeship); South-Western (Lubusz Voivodeship, Lower Silesia Voivodeship, Opole Voivodeship and Silesia Voivodeship); South-Eastern (Swiętokrzyskie, Lesser Poland and Podkarpackie) (Figure 2), corresponding to the distribution of schools and children aged 7–12.

This information is in Protocol study published in Nutrients Journal in article Hamulka et al. (2024).

Hamulka, J.; Czarniecka-Skubina, E.; Gutkowska, K.; Drywień, M.E.; Jeruszka-Bielak, M. Nutrition-Related Knowledge, Diet Quality, Lifestyle, and Body Composition of 7–12-Years-Old Polish Students: Study Protocol of National Educational Project Junior-Edu-Żywienie (JEŻ). Nutrients 2024, 16, 4. https://doi.org/10.3390/nu16010004.

Taking into account various socioeconomic groups and geographical characteristics of the group allowed to obtain a reliable picture of the knowledge and consumption of milk and dairy products among primary school students aged 10-12, but only at this moment in time. In a longitudinal study you repeatedly collect data from the same sample over an extended period of time, but sometimes it is difficult to have the entire sample at all the time. We will continue our research in a longitudinal study.

As suggested, we improved discussion by adding more information about the nature of cross-sectional research and correlative relationships between the studied factors and dairy consumption

This was given in the limitation, but now in the revised manuscript we have additionally supplemented this part with a broader explanation the nature of cross-sectional study and suggestions for further research.

Comments 2:

  1. Improved Documentation of the Questionnaire:
  • Questionnaire Details: Include specific examples of questionnaire questions and explain how it was validated. This could help other researchers replicate the study and verify the reliability and validity of the instrument.
  • Validation Methodology: Provide more details on the validation process of the questionnaire used to ensure that the responses are representative and accurate.

Response 2:

Thank you for comments.

To describe the teenagers’ diets, we used a validated tool dedicated to Polish teenagers, SF-FFQ4PolishChildren® was used. The questionnaire validation process was described in detail in Kowalkowska et al [2019] - reference 23 in our manuscript. Moreover, in the Study Protocol of National Educational Project Junior-Edu-Żywienie (JEŻ) (doi: 10.3390/nu16010004 - reference 23) we described in detail all the tools used in this project.

Additionally, the questionnaires were checked in pilot studies conducted in all study groups (children, parents), residing in all three environments, i.e., village, town, and city.

There is evidence of many advantages of using brief dietary tools (FFQ), although several shortcomings have been reported, e.g., the data relies heavily on memory, therefore declining cognitive ability may result in errors when reporting on food frequency consumption; the food list cannot cover all the foods consumed by respondents, which may lead to underreporting; the use of a questionnaire causes some uncertainty due to social desirability bias, especially in females who may provide more socially acceptable answers [Klesges et al., 2004].  However, these are tools that are commonly used and recommended in nutrition assessment [Cade et al., 2004; Golley et al., 2017; FAO, 2018]

  • Klesges, L.M.; Baranowski, T.; Beech, B.; Cullen, K.; Murray, D.M.; Rochon, J.; Pratt, C. Social desirability bias in self-reported dietary, physical activity and weight concerns measures in 8- to 10-year-old African-American girls: Results from the Girls Health Enrichment Multisite Studies (GEMS). Prev. Med. 2004, 38, 78–87. [Google Scholar] [CrossRef] [PubMed]
  • Food and Agriculture Organization. Dietary Assessment: A Resource Guide to Method Selection and Application in Low Resource Settings. 2018. Available online: http://www.fao.org/3/i9940en/I9940EN
  • Cade J, Burley V, Warm D, Thompson R, Margetts B. (2004) Food-frequency questionnaires: a review of their design, validation and utilisation. Nutr Res Rev., 17(1): 5–22
  • Golley RK, Bell LK, Hendrie GA, Rangan AM, Spence A, McNaughton SA, et al. (2017): Validity of short food questionnaire items to measure intake in children and adolescents: a systematic review. J Hum Nutr Diet, 30(1): 36–50.
  • Kowalkowska, J.; Wadolowska, L.; Hamulka, J.; Wojtas, N.; Czlapka-Matyasik, M.; Kozirok, W.; Bronkowska, M.; Sadowska, J.; Naliwajko, S.; Dziaduch, I.; et al. Reproducibility of a Short-Form, Multicomponent Dietary Questionnaire to Assess Food Frequency Consumption, Nutrition Knowledge, and Lifestyle (SF-FFQ4PolishChildren) in Polish Children and Adolescents. Nutrients 2019, 11, 2929. doi:10.3390/nu11122929

In addition, in our manuscript in section 2.2. Data collection, we have described detailed questions and proposed answers. The SF-FFQ4PolishChildren® questionnaire is available in Polish and English versions in the publication of Hamulka et al. (2018) - supplementary (https://doi.org/10.3390/nu10101439).

We strongly encourage you to use our proven research tool (Therefore, we encourage you to use our proven research tool, we will be very pleased).

  • Hamulka, J.; Wadolowska, L.; Hoffmann, M.; Kowalkowska, J.; Gutkowska, K. Effect of an Education Program on Nutrition Knowledge, Attitudes toward Nutrition, Diet Quality, Lifestyle, and Body Composition in Polish Teenagers. The ABC of Healthy Eating Project: Design, Protocol, and Methodology. Nutrients 2018, 10, 1439. https://doi.org/10.3390/nu10101439

Comments 3:

  1. Address the discrepancy between parents' and children's perceptions, offering possible explanations or proposing methods for future research to minimize this variability.
  • Comparative Analysis: Conduct a deeper analysis of why there is a significant discrepancy between parents' and children's perceptions of the frequency of dairy consumption.
  • Proposed Solutions: Suggest methods for future research that could minimize this discrepancy, such as using mixed methods that include both self-reporting and direct observations.

Response 3:

Thank you for this comment and for bringing this to attention. We concur with your suggestion.

Thank you for this comment and for pointing this out. “Future research should use mixed methods that including both self-report and direct observations to minimize discrepancies between parents' and children's perceptions”. However, this may be difficult in very large cohort studies.
However, we would like to point out that, the discrepancies between parents and children result from the fact that children see the world differently than their parents. Children completed surveys at school and in the classroom, while parents often answered about the actual consumption of milk and dairy products by parents. Separating the research into parents and children according to the authors allowed us to avoid influencing the responses of one group (children) to the other (parents). Neither group could agree on the answer among themselves. This result could be disturbed if they completed the questionnaire together, by prompting the children to answer the questions by their parents, and this is how we know the actual state of knowledge of the children and their parents, as well as the frequency of consumption of these products by the children. In our opinion, these differences cannot be eliminated. This method of conducting research facilitates (realizes) the interpretation of the obtained results and allows for the inclusion of further educational activities aimed at both children and their parents.

Comments 4:

  1. Inclusion of Longitudinal Analyses:
  • Recommendation for Future Studies: Suggest conducting longitudinal studies that can follow the same subjects over time to establish more robust causal relationships between the studied factors and dairy consumption.
  • Exploration of Temporal Trends: Discuss the possibility of long-term follow-up of the current sample to observe how consumption patterns and associated factors change over time.

Response 4:

Thank you very much for this suggestion.

We have added relevant information in the implication:

"Future studies should consider using conducting longitudinal studies that can follow the same subjects over time to establish more robust causal relationships between the studied factors and dairy consumption".

Comments 5:

  1. Expansion of International Context:
  • International Comparisons: Include comparisons with similar studies conducted in other countries to place the results in a global context. This could help identify specific cultural factors influencing consumption patterns.
  • Global Relevance: Discuss how the findings could be applicable or relevant to populations outside of Poland.

Response 5:

Thank you for your suggestion. There are few studies in the available literature on trends in the consumption of milk and dairy products after 2015 in this age group in our region.

Our results are discussed in lines 369-387.

Our observations suggest that similar implications could be applied in Central-Northern Europe, which is characterized by similar dietary patterns.

Comments 6:

  1. Deepening the Analysis of Socioeconomic Factors:
  • Socioeconomic Breakdown: Conduct a more detailed analysis of how different socioeconomic levels influence dairy consumption, as this factor can be crucial for the implementation of effective public policies.

Response 6:

Thank you for your suggestion.

In our in our analyzes income in the case of milk and dairy products was not taken into account because children in the Polish countryside usually have greater access to these products from their own farms than children from cities, despite lower household incomes. In turn, in cities, it has been fashionable for many years to consume lactose-free products and avoid milk and dairy products, despite higher family incomes. Based on the research analyses and the literature review [Wiza, 2022], it was shown that changes in the level of milk consumption were small in relation to changes in prices expressed in current prices, as well as consumer income. In contrast, there were changes in dairy consumption. Between 2004 and 2020, the consumption of highly processed dairy products, i.e. yoghurts, cheeses, cottage cheese, increased, while the consumption of liquid milk decreased and the consumption of fat products, i.e. butter and cream, increased at a lower rate. Therefore, analyzing the data in this respect would not be justified.

Wiza P., Consumption of milk and milk products in Poland in the years 2004-2020. Technological Progress in food processing (Postępy technolo przetwórstwa spożywczego), 2022, 2, 5-10.

All studied socio-demographic factors were used in statistical analyzes and presented in this manuscript.

Future research could consider other factors such as parents' education, family income, availability of pocket money, school program (e.g. school milk program), which would be crucial for the implementation of effective public policies.

  • Specific Interventions: Propose specific interventions aimed at various socioeconomic groups to improve dairy consumption equitably.

Response 6:

Proposals for specific interventions aimed at various socioeconomic groups to improve dairy consumption equitably depends on many factors, mainly age and health factors. Increasing the consumption of milk and dairy products can be achieved through government subsidies/actions/programs, e.g. "Program for schools". However, this should be implemented alongside nutritional education.

Moreover, education should include both the family environment (children, parents, guardians), the  school environment (pupils, teachers, collective catering - canteen, shop, vending machine) and peers (colleagues, friends).

Comments 7:

  1. Improvement in Presentation of Results:
  • Data Visualization: Use more visual and less text-dense charts and tables to present the results more clearly and comprehensibly.
  • Summary of Key Findings: Include a visual summary or infographic highlighting the study’s key findings to facilitate quick understanding of the most important points.

Response 7:

Thank you for pointing this out. However, we believe that the tables are clear and have been produced in accordance with the Journal's guidelines. To further improve their clarity, we have made minor changes to Tables 2 and 3. We hope you will find them satisfactory.

As suggested, we have prepared a graphic abstract showing the summary of the key findings.

Comments 8:

  1. Exploration of Family Habits:
  • Impact of Family Meals: Investigate more deeply how the frequency and quality of family meals affect the consumption of dairy products and other healthy foods.
  • Educational Interventions: Propose educational programs involving the whole family to promote healthy eating habits from an early age.

Response 8:

Thank you for this comment and for bringing this to attention.

Our previous qualitative research using a Focus Group Interview (FGI) shows that family meals are important in the opinion of both parents and children.

Moreover, an important aspect was preparing and eating meals together. According to literature data the benefits of eating meals together, especially at home with the family, concern psychosocial areas (lower risk of eating disorders) (Glanz et al., 2021, Langdon-Daly et al., 2017) as well as healthier eating patterns (e.g. higher consumption of fruit and vegetables, milk and products, and lower consumption of sugar-sweetened beverages (Fulkerson et al., 2014). This was also confirmed by qualitative research carried out as part of the Junior-Edu-Żywienie (JEŻ) project.

  • Czarniecka-Skubina, E.; Gutkowska, K.; Hamulka, J. The Family Environment as a Source for Creating the Dietary Attitudes of Primary School Students—A Focus Group Interview: The Junior-Edu-Żywienie (JEŻ) Project. Nutrients 2023, 15, 4930. https://doi.org/10.3390/nu15234930
  • Glanz, K, Metcalfe, J.J, Folta, S.C, Brown, A, Fiese, B. Diet and Health Benefits Associated with In-Home Eating and Sharing Meals at Home: A Systematic Review. Int. J. Environ. Res. Public. Health 2021, 18(4): 1577. doi: 10.3390/ijerph18041577.
  • Fulkerson, J.A.; Larson, N.; Horning, M.; Neumark-Sztainer, D. A Review of Associations between Family or Shared Meal Frequency and Dietary and Weight Status Outcomes across the Lifespan. Journal of Nutrition Education And Behavior 2014, 46: 2–19. doi: 10.1016/j.jneb.2013.07.012.
  • Langdon-Daly, J, Serpell, L. Protective Factors against Disordered Eating in Family Systems: A Systematic Review of Research. Journal of Eating Disorders 2017, 5:12. https://doi.org/10.1186/s40337-017-0141-7

To achieve positive out- comes in educational activities related to food and nutrition, it is essential to involve children, parents, guardians, teachers, and other school staff in these efforts. This is the main goal of our Junior-Edu-Nutrition project.

Different activities are necessary to implement nutritional education for children and adolescents, whole families, and teachers. Modern IT and telecommunication tools (e.g. social media, mobile applications) should be used to support traditional methods.

Authors

Round 2

Reviewer 1 Report

Comments and Suggestions for Authors

All comments were addressed.

Author Response

Dear Reviewer,

Thank you very much for your positive response to our answers.

Best regards

Reviewer 2 Report

Comments and Suggestions for Authors The requested changes have been made correctly.

Author Response

(The authors gave the same response as above.)
